# Training Your Image Restoration Network Better with Random Weight Network as Optimization Function

**Man Zhou**[1]    **Naishan Zheng**[1,2]    **Yuan Xu**[1]    **Chunle Guo**[3]    **Chongyi Li**[3†*]

[1]S-Lab, Nanyang Technological University
[2]University of Science and Technology of China, China
[3]Nankai University, China
`man.zhou@ntu.edu.cn, lichongyi25@gmail.com`

## Abstract

The blooming progress made in deep learning-based image restoration has been largely attributed to the availability of high-quality, large-scale datasets and advanced network structures. However, optimization functions such as $\mathcal{L}_1$ and $\mathcal{L}_2$ are still de facto. In this study, we propose to investigate new optimization functions to improve image restoration performance. Our key insight is that "random weight network can be acted as a constraint for training better image restoration networks". However, not all random weight networks are suitable as constraints. We draw inspiration from Functional theory and show that alternative random weight networks should be represented in the form of a strict mathematical manifold. We explore the potential of our random weight network prototypes that satisfy this requirement: Taylor's unfolding network, invertible neural network, central difference convolution, and zero-order filtering. We investigate these prototypes from four aspects: 1) random weight strategies, 2) network architectures, 3) network depths, and 4) combinations of random weight networks. Furthermore, we devise the random weight in two variants: the weights are randomly initialized only once during the entire training procedure, and the weights are randomly initialized in each training epoch. Our approach can be directly integrated into existing networks without incurring additional training and testing computational costs. We perform extensive experiments across multiple image restoration tasks, including image denoising, low-light image enhancement, and guided image super-resolution to demonstrate the consistent performance gains achieved by our method.

## 1   Introduction

Image restoration is a challenging task that involves recovering a latent clear image from a given degraded observation. This task is highly ill-posed as there are infinite feasible results for a single degraded image [1, 2]. Researchers have tackled this problem through two main approaches: traditional optimization methods [3, 4, 5] and deep learning-based methods [2, 6, 7].

Traditional methods formulate image restoration as an optimization problem and use various image priors to regularize the solution space of the latent clear image, such as low-rank prior [4, 5], dark channel prior [8, 9, 10], graph-based prior [11, 12], total variation regularization [13, 14], and sparse image priors [15, 16]. However, these priors require carefully designed priors and involve iteration optimization, making them computationally expensive.

In recent years, deep neural networks (DNNs) have shown promising results in image restoration tasks [17, 18, 19]. These methods have three key components: data, model, and optimization function.

---

*† corresponding author.

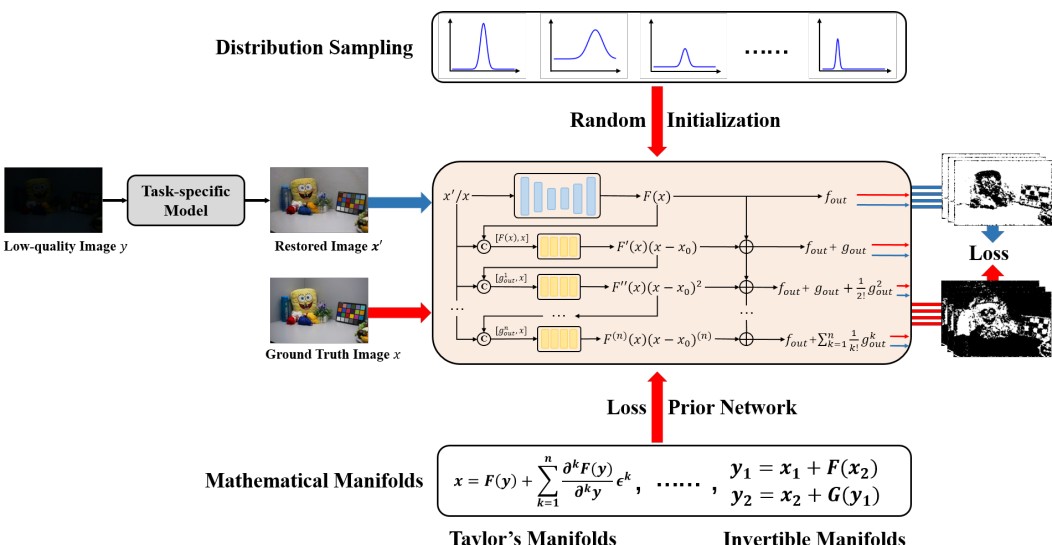

Figure 1: The flowchart of **"Random Weight Networks as Optimization Prior for Image Restoration"**. In detail, the output of the task-specific model and the ground truth are respectively fed into the random weight network to constrain the restoration network training, where the random weight network is required to satisfy a strict mathematical manifold, and its weights are randomly initialized.

While significant effort has been devoted to collecting high-quality and large-scale datasets and designing advanced network structures, the $\mathcal{L}1$ and $\mathcal{L}2$ losses remain the de facto optimization functions. In this study, we explore the potential of using random weight networks as a constraint for training better image restoration networks. By incorporating a random weight network as a constraint during the training process, we aim to encourage the network to learn more robust features and produce better results. This approach may help to overcome some of the limitations of traditional and deep learning-based methods for image restoration.

**Related Work.** In terms of the prior loss function studies, the representative one [20] customized a dual regression scheme for image super-resolution task in the form of loss regularization term. This method introduces an additional constraint on low-resolution data to reduce the space of the possible image super-resolution solutions. Similar to the loss regularization function, CycleGAN framework [21, 22, 23] exploited two sets of parallel generative adversarial networks to formulate the image restoration function and the image degradation mechanism respectively where the corresponding cycle mechanism is modeled in the loss function by the form of regularization term. In addition, this work [24] explored the Range-Null space decomposition to enable the relationship between realness and data consistency, and the consistency constraint is transferred into loss function. Despite the remarkable progress, the first two have been trained in a delicate manner while the remaining one only works on the particular forms of known degradation matrix. These issues leave room to further study the potential of loss function for image restoration.

In Figure 1, we show the flowchart of a random weight network as optimization prior to train an image restoration network, where the random weight network is treated as an optimization function without incurring additional training and testing computational costs. Despite the succinct idea, there still exists a question "whether any network architecture with random weights can be used as optimization function?". With this question, we propose our analysis and solution. Specifically, we stand on the Functional theory and show that alternative random weight networks should be represented in the form of a strict mathematical manifold. We explore the potential of our random weight network prototypes that satisfy this requirement: Taylor's unfolding network, invertible neural network, central difference11 convolution, and zero-order filtering. We investigate these prototypes from four aspects: 1) random weight strategies, 2) network architectures, 3) network depths, and 4) combinations of random weight networks. Based on the above settings, we employ the random weight networks as optimization functions to better optimize the task model in the following two variants: 1) the weights are randomly initialized only once during the entire training procedure, and 2) the weights are randomly initialized in each training iteration epoch. The illustration of random

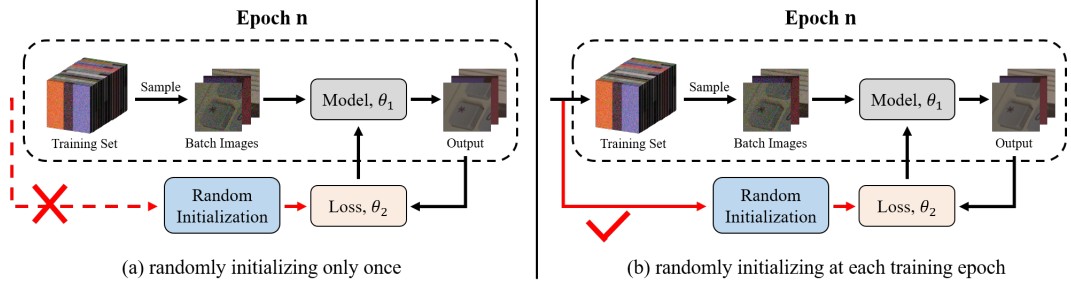

Figure 2: Random Weight Strategies: (a) the weights are randomly initialized only once during the entire training procedure; (b) the weights are randomly initialized in each training iteration epoch.

weights strategies is shown in Figure 2. Our approach is verified over the representative baselines across multiple image restoration tasks including image denoising, low-light image enhancement, and guided image super-resolution and the extensive experimental results demonstrate its effectiveness.

To summarize, we make the following key contributions. 1) It is the first attempt to propose the insight that "random weight network can be acted as a constraint for training better image restoration networks". Our insight would spark the studies of optimization functions for image restoration. 2) Orthogonal to the existing data and model studies conducted in the field of deep learning-based image restoration, our proposed approach is a plug-and-play, thus improving image restoration performance without changing the original model and data configurations. 3) Our approach can improve the performance of multiple image restoration tasks without incurring additional training and testing computational costs.

## 2 Methodology

In this section, we will first introduce the prior image restoration optimization flowchart and discuss the limitation of current optimization functions. We then detail the alternative random weight networks as optimization functions and show their feasibility. At last, we present the random weights initialization strategies.

### 2.1 Image Restoration Flowchart

Suppose that the restoration task model as $f(\mathbf{X})$ that transforms the input image $\mathbf{x}$ to the output $\mathcal{Y}$, the restoration process can be written as

$$\mathcal{Y} = f(\mathbf{X}), \tag{1}$$

Suppose that the ground truth as GT, the commonly used optimization function $e.g.,$ $\mathcal{L}_1$ or $\mathcal{L}_2$ can be written as

$$\mathbf{L} = ||\mathbf{GT} - \mathcal{Y}||_{1,2}, \tag{2}$$

where $||.||_{1,2}$ is the image-level $\mathcal{L}_1$ or $\mathcal{L}_2$ loss.

**Discussion.** From the Bayesian perspective, it is well known that minimizing $\mathcal{L}_1$ or $\mathcal{L}_2$ can be equivalent to maximum likelihood estimation in regression. The prediction of a regressor can be treated as the mean of a noisy prediction distribution, which is modeled as a Gaussian or Laplace distribution in the classic probabilistic interpretation:

$$\mathcal{L}_2 : \mathrm{p}(\mathcal{Y}|\mathbf{X}; \theta) = \mathcal{N}(\mathrm{GT}; \mathcal{Y}, \sigma_{\mathrm{noise}}^2 \mathrm{I}), \tag{3}$$
$$\mathcal{L}_1 : \mathrm{p}(\mathcal{Y}|\mathbf{X}; \theta) = \mathcal{L}(\mathrm{GT}; \mathcal{Y}, \mathrm{b}), \tag{4}$$

where $\sigma_{noise}$ is the scale of an i.i.d. error term $\epsilon \sim \mathcal{N}(0, \sigma_{\mathrm{noise}}^2 \mathrm{I})$ and $\mathrm{b} = \sqrt{\frac{\sigma_{\mathrm{noise}}}{2}}$. However, the real distribution is more complex, making the model optimization difficult and pushing the model prediction biased. To solve this issue, we propose to customize the additional network as loss prior regularization term to better constrain the output close to the ground truth distribution.

## 2.2 Random Weight Network Manifold

In this study, we found that only random weight networks adhering to a specific mathematical manifold are suitable as optimization functions. To meet this requirement, we propose alternative random weight networks: Taylor's Unfolding Network, Invertible Neural Network, Central Difference Convolution, and Zero-order Filtering.

**Taylor's Unfolding Network Manifold in Figure 3.** Drawing inspiration from image decomposition, we leverage Taylor's unfolding [25] to formulate the manifold that is oriented towards decomposition

$$\mathcal{Y} = \mathcal{T}X + N, \tag{5}$$

where we denote the observation, latent clear image, transformation matrix, and error term as $\mathcal{Y}$, X, $\mathcal{T}$, and N, respectively.

Let $\mathcal{Y}_0 = \mathcal{T}X = \mathcal{Y} - N$, we learn X by the function $\mathcal{F}$

$$X = \mathcal{F}(\mathcal{Y}_0) = \mathcal{F}(\mathcal{Y} - N). \tag{6}$$

When only regarding $n$ order Taylor's approximations, it can be simplified as

$$X = \mathcal{F}(\mathcal{Y}) + \sum_{k=1}^{n} \frac{\partial^k \mathcal{F}(\mathcal{Y})}{\partial^k \mathcal{Y}} \epsilon^k. \tag{7}$$

Recalling Equation (7), for the $k$ order derivative part, it can be written as

$$\mathcal{F}^k(\mathcal{Y})\epsilon^k = \frac{\partial^k \mathcal{F}(\mathcal{Y})}{\partial^k \mathcal{Y}} \epsilon^k, \tag{8}$$

To achieve this objective, we employ a Derivative function sub-network denoted as $\boldsymbol{G}$, which operates in the aforementioned process. The $k$th order output of network $\boldsymbol{G}$ is represented as $\mathcal{F}^k(\mathcal{Y})\epsilon^k$ and conveniently recorded as $g_{out}^k$

$$g_{out}^{k+1} = \boldsymbol{G}(g_{out}^k) + k \cdot g_{out}^k. \tag{9}$$

By combining the aforementioned two operational steps, we can obtain the final output of the $n$th order deep Taylor's approximations framework as follows:

$$O = \mathcal{F}(\mathcal{Y}) + \sum_{k=1}^{n} \frac{1}{k!} g_{out}^k. \tag{10}$$

**Invertible Neural Network Manifold in Figure 3.** Drawing inspiration from image transformation techniques such as Fourier transform and wavelet transform, we can designate the aforementioned invertible transformation as:

$$\mathcal{Y} = \mathcal{T}X, \tag{11}$$

where $\mathcal{T}$ is the wavelet function for wavelet transform while representing the Trigonometric basis for Fourier transform.

To capture the general invertible manifold, the fundamental units divide the input into $x_1$ and $x_2$. Reversible blocks receive $(X_1, X_2)$ and generate outputs $(\mathcal{Y}_1, \mathcal{Y}_2)$ using additive coupling rules inspired by NICE's transformation [26, 27, 28], as demonstrated in Figure 3(right):

$$\mathcal{Y}_1 = X_1 + \mathcal{F}(X_2),$$
$$\mathcal{Y}_2 = X_2 + \mathcal{G}(\mathcal{Y}_1). \tag{12}$$

**Central Difference Convolution Manifold in Figure 4.** In deep networks, the vanilla 2D convolution is a fundamental operator with two steps: 1) *sampling*, selecting a local neighbor region $\mathcal{R}$ from input feature map $x$; and 2) *aggregating*, combining sampled values with learnable weights $w$. This process formulates the output feature map $\mathcal{Y}$

$$\mathcal{Y}(p_0) = \sum_{p_n \in \mathcal{R}} w(p_n) \cdot X(p_0 + p_n), \tag{13}$$

Here, $p_0$ represents the current location on both input and output feature maps, while $p_n$ iterates over locations in $\mathcal{R}$. For example, in a convolution operator with a 3×3 kernel and dilation 1, the local receptive field region is defined as $\mathcal{R} = \{(-1, -1), (-1, 0), \cdots, (0, 1), (1, 1)\}$.

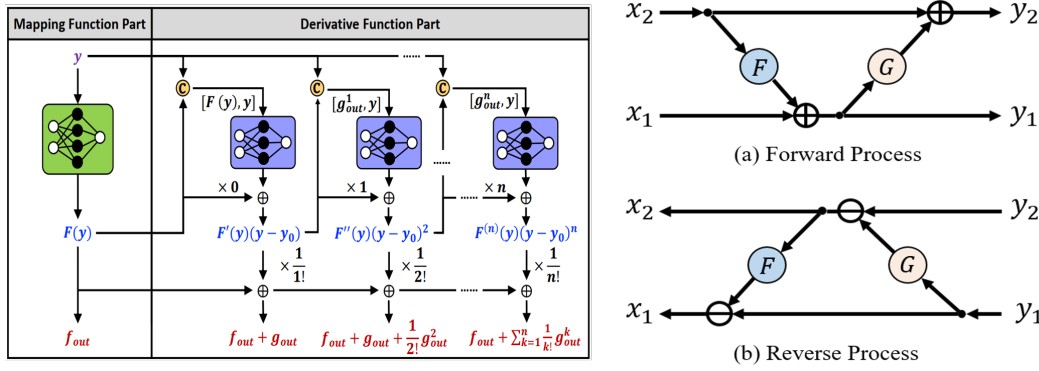

**Taylor Framework's manifold**      **Invertible Network's manifold**

Figure 3: The functional manifolds of deep Taylor's approximations framework (left) and invertible neural network (right).

Contrary to the vanilla convolution, the central difference convolution incorporates central gradient features to augment the representation and generalization capabilities. This can be formulated as:

$$\mathcal{Y}(p_0) = \sum_{p_n \in \mathcal{R}} w(p_n) \cdot (\mathrm{X}(p_0 + p_n) - \mathrm{X}(p_0)). \tag{14}$$

**Reverse Filtering Network Manifold inspired by [29] in Figure 4.** Definition. Let $(\mathcal{H}, d)$ be a metric space, and $T : \mathcal{H} \to \mathcal{H}$ be a mapping function. For all $x, y \in \mathcal{H}$, if there exists a constant $c \in [0, 1)$ such that the following formula holds:

$$d(T(x), T(y)) \leq c \cdot d(x, y), \tag{15}$$

mapping $T : \mathcal{H} \to \mathcal{H}$ is called Contraction Mapping.

*Theorem-1. Given a function $\Phi : \mathcal{H} \to \mathcal{H}$, a variable $x^*$ is considered a fixed point if $\Phi(x^*) = x^*$. If $\Phi$ is a contraction mapping, it guarantees the existence of a unique fixed point $x^*$ in $\mathcal{H}$. Moreover, the fixed point $x^*$ can be determined using the following iterative process. Let the initial guess be $x_0$ and define a sequence $\{x_n\}$ such that $x_n = \Phi(x_{n-1})$. As the iterative process converges, we have $\lim_{n \to \infty} x_n = x^*$.*

Reverse Filtering. The function $\mathcal{F}(\cdot)$ can be viewed as a set of versatile filters used for image smoothing. This filtering process can be described as $\mathcal{Y} = \mathcal{F}(\mathrm{X})$, where X represents the input image and $\mathcal{Y}$ represents the filtering result. By performing reverse filtering, we can estimate X without explicitly computing $\mathcal{F}^{-1}(\cdot)$ and update the restored image based on the filtering effect, resulting in:

$$\mathrm{X}^{k+1} = \mathrm{X}^k + \mathcal{Y} - \mathcal{F}(\mathrm{X}^k), \tag{16}$$

Here, $\mathrm{X}^k$ represents the current estimation of X in the *k*-th iteration. The iteration begins with $\mathrm{X}^0 = \mathcal{Y}$, and as the iteration count $k$ increases, $\mathrm{X}^k$ gradually approaches the true X. To facilitate this process, we introduce the auxiliary function $\varphi(\cdot)$ as: $\varphi(\mathrm{X}) = \mathrm{X} + \mathcal{Y} - \mathcal{F}(\mathrm{X})$. Therefore, the above iterative process can be regarded as a fixed point iteration

$$\mathrm{X}^{k+1} = \varphi(\mathrm{X}^k). \tag{17}$$

In Figure 4(right), the filtering process $\mathcal{F}(\cdot)$ is realized using the Multi-scale Gaussian Convolution Module, which satisfies the necessary condition outlined in Theorem-1. Specifically, taking $\varphi_1(\cdot)$ as an example, the sufficient condition for Theorem 1 to hold is that $\varphi_1(\mathbf{H})$ constitutes a contraction mapping

$$\begin{aligned}
&\|\varphi_1(\mathbf{H}_a) - \varphi_1(\mathbf{H}_b)\| \\
&= \left\| \left[ \mathbf{H}_a + \hat{\mathbf{L}} - f(\mathbf{H}_a) \right] - \left[ \mathbf{H}_b + \hat{\mathbf{L}} - f(\mathbf{H}_b) \right] \right\| \\
&= \|[\mathbf{H}_a - f(\mathbf{H}_a)] - [\mathbf{H}_b - f(\mathbf{H}_b)]\| \leq c \cdot \|\mathbf{H}_a - \mathbf{H}_b\|.
\end{aligned} \tag{18}$$

For linear filters, the condition is further simplified as

$$\|\mathbf{H} - f(\mathbf{H})\| \leq c \cdot \|\mathbf{H}\|. \quad c \in [0, 1). \tag{19}$$

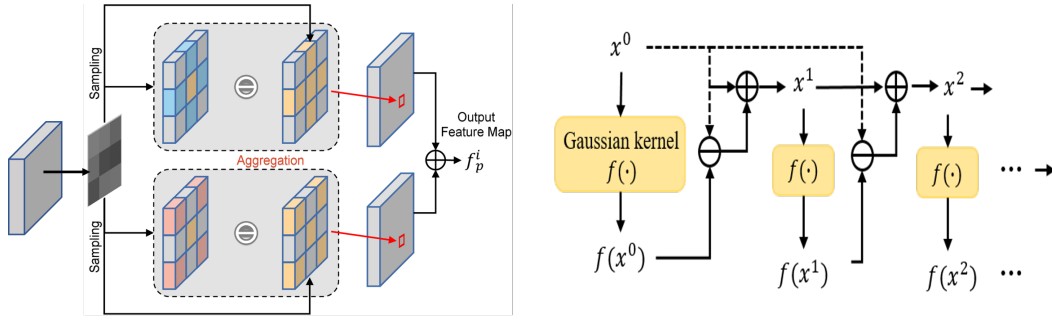

**Central Difference's manifold**                    **Reverse Filtering's manifold**

Figure 4: The functional manifolds of deep central difference convolution (left) and reverse filtering network (right).

## 2.3 Random Weight Strategies

Based on the above settings, we employ random weight networks as optimization functions to better optimize the image restoration models. In Figure 2, we present the following two variants for the random weights initialization:

- (1) the weights are randomly initialized only once during the entire training procedure,
- (2) the weights are randomly initialized in each training iteration epoch, denoted as "epochR";

We also discuss their effects in the experiment part.

## 2.4 Optimization Pipeline

Suppose that the randomly initialized manifold model as $f_{random}(.)$, it is employed as the complementary loss function to the original image-level loss function. The total loss function is remarked as

$$\mathbf{L} = ||\mathbf{GT} - \mathbf{y}||_{1,2} + \lambda||f_{random}(\mathbf{GT}) - f_{random}(\mathbf{y})||_{1,2}, \tag{20}$$

where $\lambda$ indicates the weighted factor, $||.||_{1,2}$ is the $\mathcal{L}_1$ or $\mathcal{L}_2$ loss, and $\mathbf{GT}$ denotes the ground truth.

# 3 Experiments

To demonstrate the efficacy of our proposed approach, we conduct extensive experiments on multiple image restoration tasks, including image denoising, low-light image enhancement, and guided image super-resolution. *We provide more experimental results in the Appendix.*

## 3.1 Experimental Settings

**Image Enhancement.** We verify our approach on the image enhancement benchmarks, LOL [30]. Further, we adopt representative SID [31] and DRBN [32] as two baselines.

**Image Denoising.** Following [33], we employ the widely-used SIDD dataset [34] as the training benchmark. Further, the corresponding performance evaluation is conducted on the remaining validation samples from the SIDD dataset. Two representative image denoising algorithms DnCNN [35] and MPRnet [33] are selected as the baselines.

**Guided Image Super-resolution.** Following [7, 36], we adopt the pan-sharpening, the representative task of guided image super-resolution, for evaluations. The WorldView II and GaoFen2 datasets [7, 36] are used for experiments. We employ INNformer [7] and SFINet [36] as the baselines.

Several widely-used image quality assessment (IQA) metrics are employed to evaluate the performance, including the relative dimensionless global error in synthesis (ERGAS) [37], PSNR, the spectral angle mapper (SAM) [38], SSIM.

Table 1: Quantitative comparison of different random weight networks with different random weight initialization strategies on image de-noising task.

| Model | Configurations | SIDD | |
|---|---|---|---|
| | | PSNR | SSIM |
| | Original | 37.1992 | 0.8954 |
| DnCNN | +Taylor | 37.3163 | 0.8955 |
| | +Taylor+epochR | 37.3719 | 0.8954 |
| | +CDC | 37.2329 | 0.8958 |
| | +CDC+epochR | 37.2784 | 0.8955 |
| | +INN | 37.3168 | 0.8970 |
| | +INN+epochR | 37.3318 | 0.8964 |
| | +Reverse | 37.3162 | 0.8965 |
| | +Reverse+epochR | 37.3321 | 0.8955 |
| | Original | 39.2372 | 0.9159 |
| MPRnet | +Taylor | 39.2953 | 0.9161 |
| | +Taylor+epochR | 39.3283 | 0.9161 |
| | +CDC | 39.2609 | 0.9160 |
| | +CDC+epochR | 39.2821 | 0.9161 |
| | +INN | 39.2729 | 0.9162 |
| | +INN+epochR | 39.3317 | 0.9162 |
| | +Reverse | 39.2446 | 0.9160 |
| | +Reverse+epochR | 39.2660 | 0.9161 |

Table 2: Quantitative comparison of different random weight networks with different random weight initialization strategies on image enhancement task.

| Model | Configurations | LOL | |
|---|---|---|---|
| | | PSNR | SSIM |
| | Original | 20.2461 | 0.7920 |
| SID | +Taylor | 20.5864 | 0.7971 |
| | +Taylor+epochR | 20.6018 | 0.7975 |
| | +CDC | 20.3298 | 0.7927 |
| | +CDC+epochR | 20.4750 | 0.7999 |
| | +INN | 20.3178 | 0.7944 |
| | +INN+epochR | 20.3958 | 0.7924 |
| | +Reverse | 20.5014 | 0.7941 |
| | +Reverse+epochR | 20.5203 | 0.7943 |
| | Original | 19.8509 | 0.7769 |
| DRBN | +Taylor | 20.1156 | 0.7778 |
| | +Taylor+epochR | 20.2405 | 0.7791 |
| | +CDC | 19.7952 | 0.7851 |
| | +CDC+epochR | 20.0756 | 0.7837 |
| | +INN | 19.8543 | 0.7774 |
| | +INN+epochR | 20.1913 | 0.7769 |
| | +Reverse | 19.9547 | 0.7765 |
| | +Reverse+epochR | 20.1358 | 0.7751 |

Table 3: Quantitative comparison of different random weight networks on the guided image super-resolution task.

| Model | Configurations | WorldView-II | | | | GaoFen2 | | | |
|---|---|---|---|---|---|---|---|---|---|
| | | PSNR↑ | SSIM↑ | SAM↓ | ERGAS↓ | PSNR↑ | SSIM↑ | SAM↓ | EGAS↓ |
| | Original | 41.6903 | 0.9704 | 0.0227 | 0.9514 | 47.3528 | 0.9893 | 0.0102 | 0.5479 |
| INNformer | +Taylor | 41.8168 | 0.9716 | 0.0224 | 0.9276 | 47.4058 | 0.9901 | 0.0101 | 0.5356 |
| | +CDC | 41.8072 | 0.9715 | 0.0224 | 0.9276 | 47.4121 | 0.9902 | 0.0100 | 0.5354 |
| | +INN | 41.8229 | 0.9717 | 0.0223 | 0.9276 | 47.4233 | 0.9904 | 0.0100 | 0.5353 |
| | +Reverse | 41.7293 | 0.9711 | 0.0226 | 0.9276 | 47.4010 | 0.9901 | 0.0101 | 0.5354 |
| | Original | 41.7244 | 0.9725 | 0.0220 | 0.9506 | 47.4712 | 0.9901 | 0.0102 | 0.5462 |
| SFINet | +Taylor | 41.9314 | 0.9723 | 0.0219 | 0.9278 | 47.6132 | 0.9911 | 0.0101 | 0.5277 |
| | +CDC | 41.8943 | 0.9719 | 0.0220 | 0.9283 | 47.5990 | 0.9910 | 0.0101 | 0.5281 |
| | +INN | 41.9521 | 0.9727 | 0.0217 | 0.9278 | 47.6316 | 0.9916 | 0.0101 | 0.5275 |
| | +Reverse | 41.9217 | 0.9722 | 0.0218 | 0.9281 | 47.6227 | 0.9914 | 0.0101 | 0.5275 |

Table 4: Ablation study of the impact of network architecture on image denoising task.

| Model | Configurations | SIDD | |
|---|---|---|---|
| | | PSNR↑ | SSIM↑ |
| | Original | 37.1992 | 0.8954 |
| DnCNN | +Taylor+epochR | 37.3719 | 0.8954 |
| | +Taylor+epochR+Transformer | 37.3560 | 0.8958 |
| | +INN+epochR | 37.3318 | 0.8964 |
| | +INN+epochR+Transformer | 37.3297 | 0.8961 |
| | Original | 39.2372 | 0.9159 |
| MPRnet | +Taylor+epochR | 39.3283 | 0.9161 |
| | +Taylor+epochR+Transformer | 39.2783 | 0.9160 |
| | +INN+epochR | 39.3317 | 0.9162 |
| | +INN+epochR+Transformer | 39.2756 | 0.9159 |

Table 5: Ablation study of the impact of network architecture on image enhancement task.

| Model | Configurations | LoL | |
|---|---|---|---|
| | | PSNR | SSIM |
| | Original | 20.2461 | 0.7920 |
| SID | +Taylor+epochR | 20.6018 | 0.7975 |
| | +Taylor+epochR+Transformer | 20.5864 | 0.7971 |
| | +INN+epochR | 20.3958 | 0.7924 |
| | +INN+epochR+Transformer | 20.3178 | 0.7944 |
| | Original | 19.8509 | 0.7769 |
| DRBN | +Taylor+epochR | 20.2405 | 0.7791 |
| | +Taylor+epochR+Transformer | 20.1826 | 0.7784 |
| | +INN+epochR | 20.1913 | 0.7769 |
| | +INN+epochR+Transformer | 20.1196 | 0.7772 |

## 3.2 Experimental Settings

**Image Enhancement.** We verify our approach on the image enhancement benchmarks, LOL [30]. Further, we adopt representative SID [31] and DRBN [32] as two baselines.

Table 6: Ablation study of the impact of network depth on image denoising task.

| Model | Configurations | SIDD | |
|---|---|---|---|
| | | PSNR | SSIM |
| | Original | 37.1992 | 0.8954 |
| DnCNN | +CDC+epochR | 37.2784 | 0.8955 |
| | +CDC(3)+epochR+Depth | 37.2218 | 0.8921 |
| | +CDC(7)+epochR+Depth | 37.2923 | 0.8930 |
| | +INN+epochR | 37.3218 | 0.8964 |
| | +INN(3)+epochR+Depth | 37.3213 | 0.8967 |
| | +INN(7)+epochR+Depth | 37.3142 | 0.8967 |
| | Original | 39.2372 | 0.9159 |
| MPRnet | +CDC+epochR | 39.2821 | 0.9161 |
| | +CDC(3)+epochR+Depth | 39.2814 | 0.9160 |
| | +CDC(7)+epochR+Depth | 39.2740 | 0.9161 |
| | +INN+epochR | 39.2729 | 0.9162 |
| | +INN(3)+epochR+Depth | 39.2758 | 0.9160 |
| | +INN(7)+epochR+Depth | 39.2737 | 0.9160 |

Table 7: Ablation study of the impact of network depth on image enhancement task.

| Model | Configurations | LoL | |
|---|---|---|---|
| | | PSNR | SSIM |
| | Original | 20.2461 | 0.7920 |
| SID | +CDC+epochR | 20.4750 | 0.7999 |
| | +CDC(3)+epochR+Depth | 20.3464 | 0.7915 |
| | +CDC(7)+epochR+Depth | 20.4258 | 0.7857 |
| | +INN+epochR | 20.3858 | 0.7924 |
| | +INN(3)+epochR+Depth | 20.4946 | 0.7862 |
| | +INN(7)+epochR+Depth | 20.2816 | 0.7959 |
| | Original | 19.8509 | 0.7769 |
| DRBN | +CDC+epochR | 20.0756 | 0.7837 |
| | +CDC(3)+epochR+Depth | 19.9188 | 0.7808 |
| | +CDC(7)+epochR+Depth | 19.9769 | 0.7795 |
| | +INN+epochR | 20.1913 | 0.7769 |
| | +INN(3)+epochR+Depth | 20.0330 | 0.7758 |
| | +INN(7)+epochR+Depth | 20.1153 | 0.7787 |

Table 8: Ablation study of the impact of initialization strategy on image denoising task.

| Model | Configurations | SIDD | |
|---|---|---|---|
| | | PSNR↑ | SSIM↑ |
| | Original | 37.1992 | 0.8954 |
| DnCNN | +CDC+epochR | 37.2784 | 0.8925 |
| | +CDC+epochR+xavier | 37.2567 | 0.8963 |
| | +INN+epochR | 37.3218 | 0.8964 |
| | +INN+epochR+xavier | 37.2890 | 0.8957 |
| | Original | 39.2372 | 0.9159 |
| MPRnet | +CDC+epochR | 39.2821 | 0.9161 |
| | +CDC+epochR+xavier | 39.2768 | 0.9160 |
| | +INN+epochR | 39.2729 | 0.9162 |
| | +INN+epochR+xavier | 39.2779 | 0.9160 |

Table 9: Ablation study of the impact of initialization strategy on image enhancement task.

| Model | Configurations | LoL | |
|---|---|---|---|
| | | PSNR | SSIM |
| | Original | 20.2461 | 0.7920 |
| SID | +CDC+epochR | 20.4750 | 0.7999 |
| | +CDC+epochR+xavier | 20.3271 | 0.7847 |
| | +INN+epochR | 20.3858 | 0.7924 |
| | +INN+epochR+xavier | 20.3257 | 0.7927 |
| | Original | 19.8509 | 0.7769 |
| DRBN | +CDC+epochR | 20.0756 | 0.7837 |
| | +CDC+epochR+xavier | 20.0136 | 0.7760 |
| | +INN+epochR | 20.1913 | 0.7769 |
| | +INN+epochR+xavier | 20.0948 | 0.7773 |

Table 10: Ablation studies of model numbers for image enhancement.

| Model | Configurations | LoL | | |
|---|---|---|---|---|
| | | PSNR | SSIM | NIQE |
| | Original | 20.2461 | 0.7920 | 4.1586 |
| SID | +CDC+epochR | 20.4750 | 0.7999 | 3.6636 |
| | +CDC+epochR+Number(357) | 20.4879 | 0.7991 | 3.6793 |
| | +CDC+epochR+Number(555) | 20.5424 | 0.7889 | 3.7738 |
| | +INN+epochR | 20.3858 | 0.7924 | 3.9210 |
| | +INN+epochR+Number(357) | 20.3516 | 0.7843 | 4.2365 |
| | +INN+epochR+Number(555) | 20.3316 | 0.7911 | 4.1289 |
| | Original | 19.8509 | 0.7769 | 4.7738 |
| DRBN | +CDC+epochR | 20.0756 | 0.7837 | 4.7850 |
| | +CDC+epochR+Number(357) | 20.0200 | 0.7789 | 4.6900 |
| | +CDC+epochR+Number(555) | 20.0403 | 0.7750 | 4.7060 |
| | +INN+epochR | 20.1913 | 0.7769 | 4.8067 |
| | +INN+epochR+Number(357) | 20.0510 | 0.7779 | 4.6957 |
| | +INN+epochR+Number(555) | 20.2572 | 0.7767 | 4.6169 |

Table 11: Ablation studies of model numbers for image denoising.

| Model | Configurations | SIDD | |
|---|---|---|---|
| | | PSNR↑ | SSIM↑ |
| | Original | 37.1992 | 0.8954 |
| DnCNN | +CDC+epochR | 37.2784 | 0.8925 |
| | +CDC+epochR+Number(357) | 37.4377 | 0.8969 |
| | +CDC+epochR+Number(555) | 37.3208 | 0.8948 |
| | +INN+epochR | 37.3218 | 0.8964 |
| | +INN+epochR+Number(357) | 37.3374 | 0.8937 |
| | +INN+epochR+Number(555) | 37.3581 | 0.8944 |
| | Original | 39.2372 | 0.9159 |
| MPRnet | +CDC+epochR | 39.2821 | 0.9162 |
| | +CDC+epochR+Number(357) | 39.2704 | 0.9161 |
| | +CDC+epochR+Number(555) | 39.2764 | 0.9160 |
| | +INN+epochR | 39.2729 | 0.9162 |
| | +INN+epochR+Number(357) | 39.2767 | 0.9160 |
| | +INN+epochR+Number(555) | 39.2818 | 0.9160 |

**Image Denoising.** Following [33], we employ the widely-used SIDD dataset [34] as the training benchmark. Further, the corresponding performance evaluation is conducted on the remaining validation samples from the SIDD dataset. Two representative image denoising algorithms DnCNN [35] and MPRnet [33] are selected as the baselines.

**Guided Image Super-resolution.** Following [39, 40], we adopt the pan-sharpening, the representative task of guided image super-resolution, for evaluations. The WorldView II and GaoFen2 datasets [7, 36] are used for experiments. We employ the representative INNformer [7] and SFINet [36] as the baselines.

### 3.3 Implementation Details

For concision, we denote some annotations of the proposed alternative solutions in strict mathematical manifolds before our presentation, and the implementation variants of the baselines are organized as the five configurations:

1) **Original**: the baseline with the basic loss ( $\mathcal{L}_1$ or $\mathcal{L}_2$);
2) **+Taylor**: complementing the basic loss with the Taylor's unfolding network manifold;
3) **+CDC**: complementing the basic loss with the central difference convolution manifold;
4) **+INN**: complementing the basic loss with the invertible neural network manifold;
5) **+Reverse**: complementing the basic loss with the reverse filtering network manifold.

### 3.4 Comparison and Analysis

For quantitative Comparison, we perform the model performance comparison over different configurations. The quantitative results of image denoising, low-light image enhancement, and guided image super-resolution are respectively presented in Table 1, Table 2, and 3. From the results, we can observe performance gain against the baselines across all the datasets in the corresponding tasks, suggesting the effectiveness of our approach. For example, in terms of image enhancement, the baseline DRBN with "+Taylor", "+CDC", "INN" and "Reverse" has obtained the 0.4dB, 0.3dB, 0.2dB, 0.2dB PSNR gains over LoL dataset respectively.

## 4 Ablation Studies

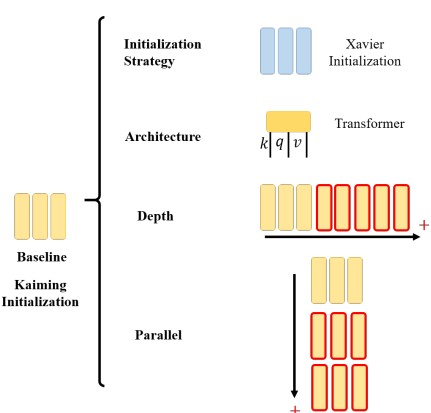

Figure 5: The ablation studies: 1) initialization strategy, 2) model architecture, 3) model depth and 4) model numbers.

To verify the stability of our approach, we conduct the following analysis from four aspects: 1) initialization strategy, 2) model architecture, 3) model depth, and 4) model numbers.

**Model architecture.** In the previous experiments, all of the random weight networks are implemented by convolution networks as default. To explore the impact of network structures, we replace the default CNN with Transformer. The results in Table 4 and Table 5 demonstrate that replacing the network rarely affects performance.

**Model depth.** For model depth, we change the model depth of the random weight network by adding more layers. To ensure a fair comparison, other factors keep the same. The results in Table 6 and Table 7 further demonstrate the stable and robust performance gains by introducing our designs.

**Initialization strategy.** In our work, the default initialization strategy is Kaiming initialization. To explore the impact of the initialization mode, we replace the default Kaiming initialization with Xavier initialization. Table 8 and Table 9 show that replacing the default initialization mode almost has little impact on the performance.

**Model numbers.** In our experiment, we use the single loss network as default and employ multiple parallel loss networks to verify the impact of model numbers. The results in Table 10 and Table 11 indicate that increasing the number of models will improve the performance. It attributes to the advantages of model ensemble.

# 5   Conclusion and Limitation

In this paper, we explore the potential of optimization functions and present our insight that "random weight networks can serve as a constraint for training improved image restoration networks." Drawing inspiration from Functional theory, we offer several alternative solutions within strict mathematical manifolds, known as "random weights network prototypes." Our approach seamlessly integrates into existing image restoration networks, and extensive experiments across multiple tasks validate its effectiveness. While our experiments are extensive, limitations in space prevent us from conducting more comprehensive evaluations, such as including experiments on image de-blurring and additional representative baselines.

## Broader Impact

Our work demonstrates the potential of random weight networks as optimization functions for low-level image restoration tasks, with applications in mobile photography, healthcare, entertainment, and other fields that rely on high-quality images. We observe no apparent negative societal consequences. While our experiments show significant performance improvements, additional real-world testing and validation are necessary to ensure robustness and generalizability.

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
