# "Training your image restoration network better with random weight network as optimization function" Supplementary Material

1 This supplementary document is organized as follows:

2 Section 0.1 provides the quantitative results for pan-sharpening.

3 Section 0.2 provides the qualitative experimental results.

4 Section 0.3 provides more provides more quantitative experimental results over ablation studies.

## 0.1 Guided Image super-resolution.

The quantitative results for pan-sharpening are summarized in Tables 1 where the best results are highlighted in bold. From the results, by integrating with our proposed random weights network by alternative mathematical manifolds, all the reported baselines have achieved consistent performance gains across all the datasets in terms of all metrics, suggesting the effectiveness of our belief.

## 0.2 Visual comparison.

Due to the page limits, the main manuscript has not presented the sufficient visual results of the reported tasks over the reported baselines. In this section, we provide the representative samples to validate the effectiveness of our belief over image de-noising task of Figure 1, Figure 2, low-light image enhancement of Figure 3. As can be seen, integrating with our belief is capable of improving the visual quality.

## 0.3 Implementation details of ablation studies.

**Initialization strategy.** In our work, the default initialization strategy is Kaiming initialization. To explore the impact of initial mode, we replace the default Kaiming initialization by Xavier initialization, reported in Table 9 and Table 8 show that replacing the default almost has little impact on performance, thus verifying the robustness of our belief.

In our experiment, we select two representative random weights network manifolds by ***Central Difference Convolution Manifold*** and ***Invertible Neural Network Manifold*** for performance verification. In detail, we employ the Xavier initialization to weight the convolution kernels within the above manifolds.

**Model architecture.** All of the loss networks are implemented by convolution network as default. To explore the architecture impact, we replace the default CNN by Transformer. The results in Table 3 and Table 2 demonstrate that replacing it rarely affects the performance.

In our experiment, we select the following random weights network manifolds by ***Taylor's Unfolding Manifold*** and ***Invertible Neural Network Manifold*** for performance verification. In detail, we replace the convolution part of main body part within Taylor's Unfolding Manifold by the transformer and the translation functions $F$ and $G$ within Invertible Neural Network Manifold by transformer.

Submitted to 37th Conference on Neural Information Processing Systems (NeurIPS 2023). Do not distribute.

Table 1: **Quantitative comparisons of guided image super-resolution.**

| Model | Configurations | WorldView-II | | | | GaoFen2 | | | |
|---|---|---|---|---|---|---|---|---|---|
| | | PSNR↑ | SSIM↑ | SAM↓ | ERGAS↓ | PSNR↑ | SSIM↑ | SAM↓ | EGAS↓ |
| | Original | 41.6903 | 0.9704 | 0.0227 | 0.9514 | 47.3528 | 0.9893 | 0.0102 | 0.5479 |
| INNformer | +Taylor | 41.8168 | 0.9716 | 0.0224 | 0.9276 | 47.4058 | 0.9901 | 0.0101 | 0.5356 |
| | +CDC | 41.8072 | 0.9715 | 0.0224 | 0.9276 | 47.4121 | 0.9902 | 0.0100 | 0.5354 |
| | +INN | 41.8229 | 0.9717 | 0.0223 | 0.9276 | 47.4233 | 0.9904 | 0.0100 | 0.5353 |
| | +Reverse | 41.7293 | 0.9711 | 0.0226 | 0.9276 | 47.4010 | 0.9901 | 0.0101 | 0.5354 |
| | Original | 41.7244 | 0.9725 | 0.0220 | 0.9506 | 47.4712 | 0.9901 | 0.0102 | 0.5462 |
| SFINet | +Taylor | 41.9314 | 0.9723 | 0.0219 | 0.9278 | 47.6132 | 0.9911 | 0.0101 | 0.5277 |
| | +CDC | 41.8943 | 0.9719 | 0.0220 | 0.9283 | 47.5990 | 0.9910 | 0.0101 | 0.5281 |
| | +INN | 41.9521 | 0.9727 | 0.0217 | 0.9278 | 47.6316 | 0.9916 | 0.0101 | 0.5275 |
| | +Reverse | 41.9217 | 0.9722 | 0.0218 | 0.9281 | 47.6227 | 0.9914 | 0.0101 | 0.5275 |

The reason is that 1) Reverse Filtering Network Manifolds have to stand on the low-pass filters for convergence maintaining where Multi-scale Gaussian Convolution Module is devised in our paper. Therefore, the architecture cannot change; 2) Central Difference Convolution Manifold is inborn with convolution architectures and thus cannot change. To this end, we select the above two samples.

**Model depth.** For model depth, we change the model depth of loss network by adding the layers. To ensure a fair comparison, the other factor keeps the same. The results in Table 5 and Table 4 demonstrate the stable performance.

In our experiment, we select two representative random weights network manifolds by ***Central Difference Convolution Manifold*** and ***Invertible Neural Network Manifold*** for performance verification. In detail, we change the default three-layer Central Difference Convolution and Invertible Neural Network by seven layers.

**Model numbers.** In our experiment, we use the single loss network as default. As shown in Table 7 and Table 6, we employ multiple parallel loss networks to verify the impact of model numbers. The results indicates that increasing the number of models will improve the performance. It attributes to the advantages of model ensemble.

In our experiment, we select two representative random weights network manifolds by ***Central Difference Convolution Manifold*** and ***Invertible Neural Network Manifold*** for performance verification. In detail, we change the default single loss network with three ones by 3-3-3 variants and 3-5-7 variants.

Table 2: **Ablation studies of model architecture for image enhancement.**

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

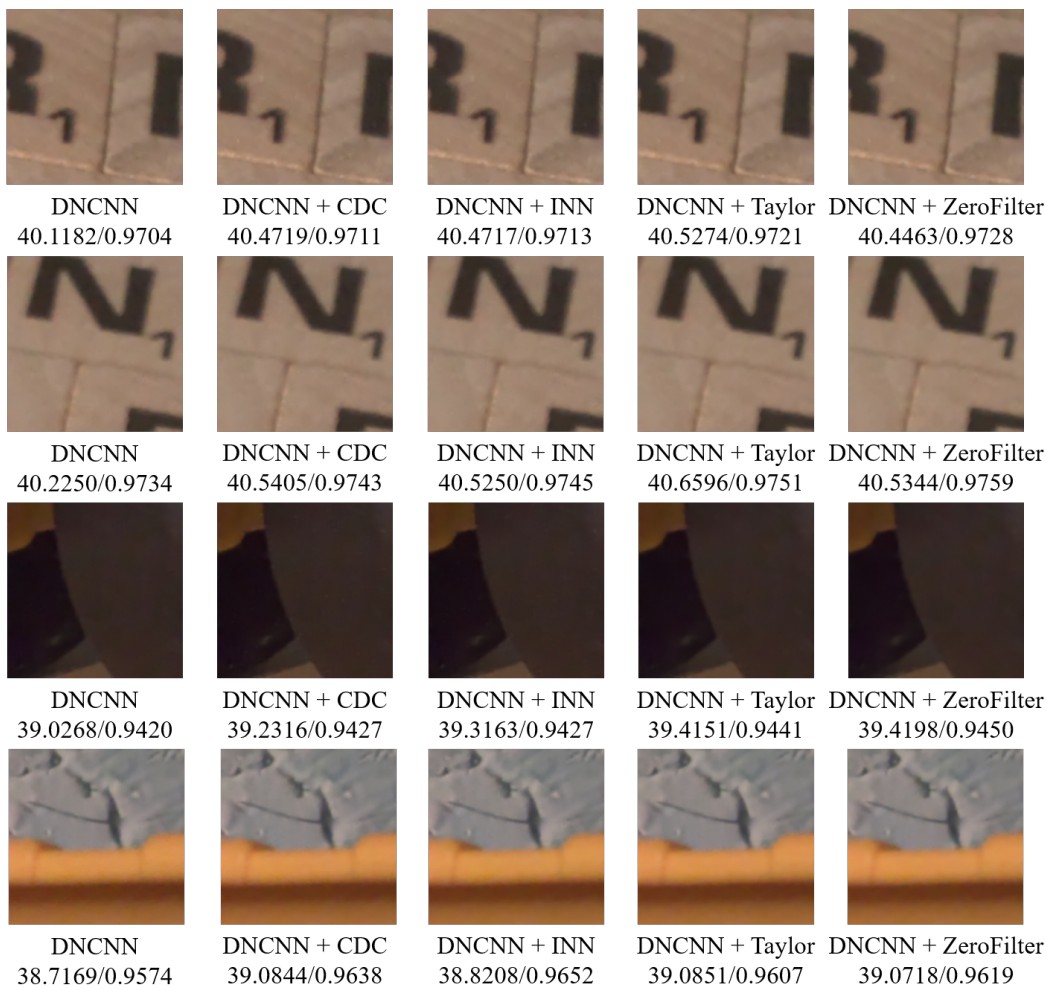

Figure 1: The visual comparison for the image de-noising. We also list the PSNR/SSIM scores under each case.

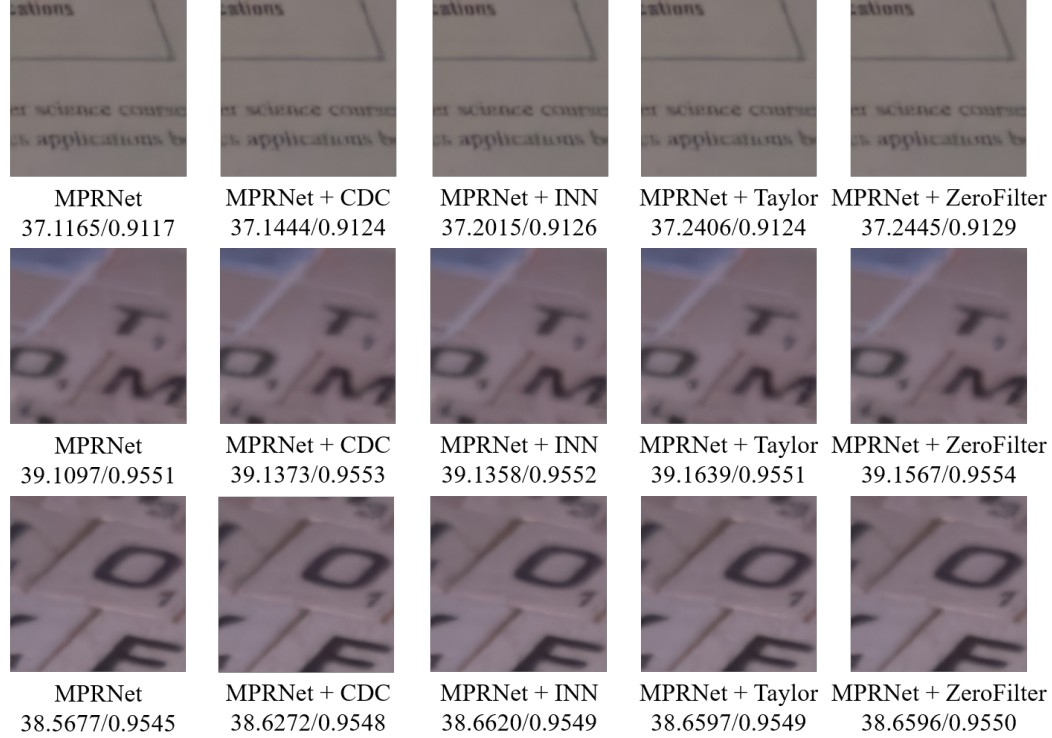

Figure 2: The visual comparison for the image de-noising. We also list the PSNR/SSIM scores under each case.

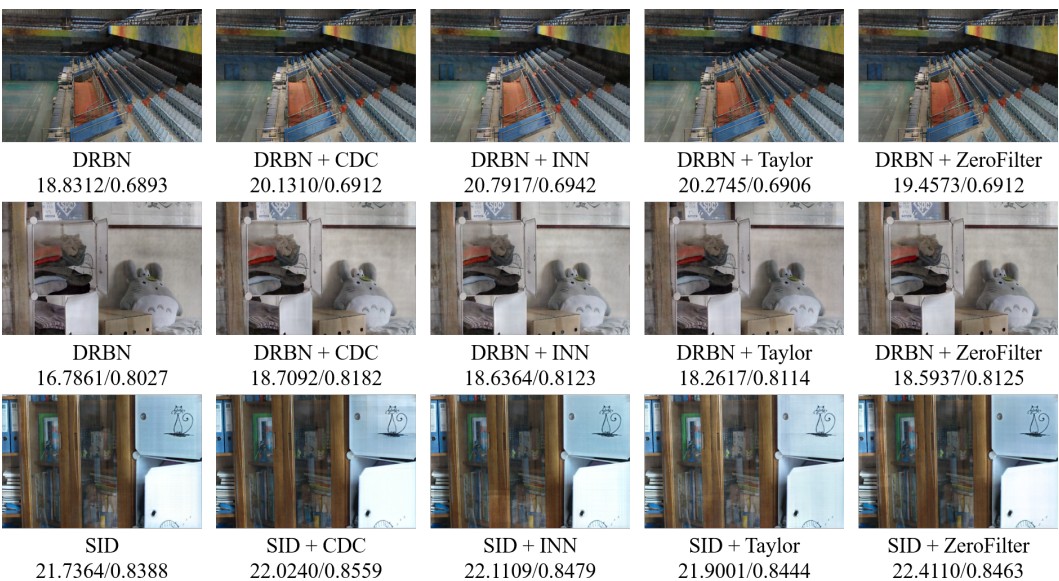

Figure 3: The visual comparison for the image enhancement. We also list the PSNR/SSIM scores under each case.