# OpenReview forum: "Training Your Image Restoration Network Better with  Random Weight Network as Optimization Function"
_NeurIPS.cc/2023/Conference — NeurIPS 2023 poster_

### Official Review · Reviewer_VxMP · 2023-07-05

**Soundness:** 4 excellent
**Presentation:** 4 excellent
**Contribution:** 3 good
**Rating:** 8
**Confidence:** 5

**Summary:**

This work introduces a novel and orthogonal approach by exploring the potential of using random weights network as a loss function. The authors have carefully designed the random weights network with theoretical constraints based on mathematical manifolds. To validate the proposed solutions, extensive experiments have been conducted on mainstream image restoration tasks. The results consistently demonstrate the effectiveness of the approach.

**Strengths:**

There are several strengths here:
1. This paper presents a pioneering exploration of the potential of using random weights network as a loss function. With a clear motivation and a series of interesting experiments, this study offers valuable insights into the applicability of the proposed approach, potentially shaping the direction of the loss function community.
2. The proposed designs seamlessly integrate into existing methods, resulting in performance improvements. A thorough set of ablation studies provides strong evidence to validate these findings.
3. The paper is well-written and maintains a high level of readability, ensuring that it is easily understandable and accessible to readers.


**Weaknesses:**

There are several weakness here:
1. To ensure clarity, it is recommended to provide detailed information regarding the experimental settings, including the specific methodologies and procedures employed. This will allow readers to have a clear understanding of how the experiments were conducted.
2. The figures and tables in the paper lack consistent style, indicating the need for a thorough review by the author to identify and correct any errors. Additionally, it would be beneficial to address potential limitations and investigate the extent to which the random weights network can be applied to various tasks or datasets.


**Questions:**

The authors are recommended to provide a detailed response addressing each concern raised in the weaknesses section.

**Limitations:**

The authors adequately addressed the limitations and potential negative societal impact of their work.

---

> ### Author Rebuttal · Authors · 2023-08-08
>
> **1,detailed information.**
>
> Thanks for pointing out this issue. Initially, due to page constraints, we've presented the relevant methodologies and procedures in the supplementary materials. Furthermore, we will share the source code to elucidate the experimental setup details. Lastly, we'll thoroughly review and elaborate on the settings in the revision to ensure readers have a clear comprehension of the conducted experiments.
>
> **2,consistent style.**
>
> Initially, we will meticulously review the entire paper, ensuring uniform style for figures and tables, while rectifying any errors. Our proposed random weights network, serving as a loss function, addresses data bias and holds theoretical applicability across various image restoration models.
>
> Following your suggestion, we expanded the application of our proposed loss functions to broader image restoration tasks, including super-resolution. Due to time constraints, we focused on the representative image super-resolution model RCAN [1] with 2-4x scaling factors to assess effectiveness. These results further validate our assertions.
>
> |setting |	RCAN |	+Taylor|	+INN|	+Zerofilter|
> |----|----|----|----|----|
> |X2	|38.27	|38.35	|38.35	|38.36|
> |X4	|32.63	|32.71	|32.69	|32.69|
>
> [1] Image super-resolution using very deep residual channel attention networks, TPAMI 2020.

---

> > ### Comment · Reviewer_VxMP · 2023-08-16
> > **Response to Authors**
> >
> > Thanks for the author's detailed reply. The rebuttal addressed my concerns. After carefully reading other reviews and the rebuttals, I agree with authors' claims on the extra training cost and differences from existed methods. Thus, I would raise my rating for this work.

---

### Official Review · Reviewer_Bde2 · 2023-07-05

**Soundness:** 4 excellent
**Presentation:** 3 good
**Contribution:** 4 excellent
**Rating:** 8
**Confidence:** 5

**Summary:**

This paper seeks to explore the untapped capabilities of random weights networks as a loss function. Inspired by mathematical manifolds, the authors propose innovative and straightforward solutions for random weights networks based on rigorous mathematical properties. Extensive experimental results across various image restoration tasks validate the efficacy of these solutions, showcasing their plug-and-play nature and ability to enhance model performance while preserving the original model and data configuration as the baseline. The novelty and interest of the idea are noteworthy.

**Strengths:**

1. Innovative approach: The authors propose a novel concept of utilizing a well-designed random weights network as a loss function, offering a plug-and-play solution that leads to remarkable performance improvements when integrated into existing baselines. This approach avoids the need for complex network architecture designs, making it highly appealing in the field of efficiency.
2. Theoretical foundation: The design of the random weights network is derived from rigorous mathematical manifolds, ensuring a solid theoretical basis. Furthermore, the authors have tailored the random sampling strategies to enrich the manifold representation, adding depth to the approach.
3. Comprehensive experiments: The paper provides extensive comparison experiments in both the main paper and the appendix, showcasing the advantages of the proposed flowchart. The inclusion of ablation studies and motivation analysis further strengthens the findings, ensuring convincing evidence of the method's effectiveness.


**Weaknesses:**

1. In all the tables, the authors have suggested to highlight the best results for a clear illustration. In addition, the more visual comparison is required to show the main body.
2. The authors have performed sufficient ablation studies. However, the corresponding experimental configuration like convolution kernel sizes needs to be detailed.
3. It would be better if the authors have presented more experimental analysis.


**Questions:**

See weaknesses part.

**Limitations:**

The authors adequately addressed the limitations and potential negative societal impact of their work.

---

> ### Author Rebuttal · Authors · 2023-08-08
>
> **1,visual comparison.**
>
> Thanks for pointing out this issue. As you suggested, we will highlight the best results for a clear illustration. In addition, we will provide the more visual comparison in the main body to enrich this work.
>
> **2,experimental configuration.**
>
> Thanks for pointing out this issue. First, due to the page limit, we have shown the corresponding experimental configuration like convolution kernel sizes of ablation studies in supplementary materials. Second, we will open the source code to highlight the details. Finally, we will check and detail the corresponding setting in revision.
>
> **3,experimental analysis.**
>
> Thanks for pointing out this issue. Due to the page limit, we have provided the partial experimental analysis and we will add the underlying working mechanism of experimental analysis to enrich this work.

---

### Official Review · Reviewer_UxgD · 2023-07-05

**Soundness:** 2 fair
**Presentation:** 3 good
**Contribution:** 2 fair
**Rating:** 4
**Confidence:** 5

**Summary:**

This paper introduces the idea that random weight networks can be used as loss functions for training image restoration networks. The paper proposes to use Taylor’s Unfolding Network, Invertible Neural Network, Central Difference Convolution, and Zero-order Filtering as random weight networks. The analysis and ablation studies show the effects of initialization strategy, model architecture, model depth, and model numbers. Experiments on image enhancement, image denoising, and guided image super-resolution validate that the proposed loss functions improve the performance of several existing image restoration methods.

**Strengths:**

+ The idea of using random weight networks as image restoration loss functions is interesting.
+ The quantitative evaluation is performed on several image restoration tasks.


**Weaknesses:**

- The paper has a major technical flaw. The abstract states that the proposed loss functions do not incur additional training computational cost. This is unreasonable because the gradients of these loss functions require additional computation cost. To be effective, the proposed loss functions must be used in conjunction with a pixel loss and are more complex than the pixel loss. It consumes more GPU memory and time during training. The paper should report the additional training cost or the extra training time.
- The quantitative improvements are not significant. From Table 1 to Table 9, the proposed loss functions have limited impact on the PSNR and SSIM results. For example, MPRNet is a representative denoising method, but its PSNR gain is less than 0.1 dB. In addition, the paper reports MPRNet achieves 39.24 dB on the SIDD dataset, which is far behind the PSNR result of 39.71 dB in the original paper of MPRNet. I suspect that this paper does not train MPRNet to convergence, and it is unreasonable to compare different loss functions without full convergence. As far as I know, NAFNet [a] achieves state-of-the-art 40.30 dB on the SIDD dataset. Are the proposed loss functions applicable to NAFNet?
- Lack of visual results in the main paper. As a paper on image restoration, it is unreasonable that the main paper does not contain any visual results. Moreover, the visual results in the supplementary materials have negligible differences, which suggests the proposed loss functions are ineffective.
- Lack of evaluation on more general image restoration tasks. The paper selects image enhancement, image denoising, and pan-sharpening as the image restoration tasks. Are the proposed loss functions applicable to more general image restoration tasks such as super-resolution?

[a] Chen et al. “Simple Baselines for Image Restoration”, ECCV, 2022.


**Questions:**

Please see the weaknesses section.

**Limitations:**

The paper does not mention any limitations of the proposed method. I believe some discussions on the perceptual quality are necessary since the quantitative improvements are limited and the visual results in the supplementary materials have negligible difference.

---

> ### Author Rebuttal · Authors · 2023-08-08
>
> **1, technical flaw.**
>
> 1) Since our proposed loss functions are used alongside pixel loss, there is no added computational burden. Pure pixel loss optimization can lead to local optima and oscillations. Conversely, our approach mitigates local oscillation and enhances model convergence. For instance, while DnCNN with pixel loss required 50K iterations to converge, integration with our designs only necessitated 30K iterations.
>
> 2) Regarding training memory, our lightweight random weights networks, detailed in the supplementary materials, ensure minimal memory overhead. Thus, employing our designs introduces only marginal and inconsequential memory demands throughout training, compared to using pure pixel loss.
>
> **2, quantitative improvements.**
>
> 1) To ensure comprehensive experimentation, we introduced a state-of-the-art model like MPRNet for validation. Given dataset limitations, the current top-performing model naturally approaches the dataset's upper bound. Despite minor impact from added efficient designs, this remains understandable.
>
> 2) Our random weights network, serving as a loss function, counters data bias and offers theoretical generality across image restoration models. Following your advice and due to time constraints, we adopted NAFNet as the baseline for evaluating our operator's efficacy. Results are provided below:
>
> |dataset | NAFNet  |    taylor  |   INN   |   zerofilter |
> |----|----|----|----|----|
> |SIDD     |      40.30|       40.41  |   40.38  |    40.46|
>
> **3, visual results.**
>
> Regarding qualitative outcomes, we'll heed your suggestion to include an increased number of visual results in the main paper.
>
> **4,  general image restoration tasks.**
>
> Following your suggestion, we expanded the application of our proposed loss functions to broader image restoration tasks, including super-resolution. Due to time constraints, we focused on the representative image super-resolution model RCAN [1] with 2-4x scaling factors to assess effectiveness. These results further validate our assertions.
>
> |setting|  RCAN  |    +Taylor |    +INN  |    +Zerofilter |
> |----|----|----|----|----|
> |X2       |     38.27    |     38.35   |  38.35  |    38.36|
> |X4      |       32.63    |     32.71   |  32.69    |  32.69|
>
> [1] Image super-resolution using very deep residual channel attention networks, TPAMI 2020.

---

> ### Comment · Area_Chair_u3Bp · 2023-08-21
>
> Dear Reviewer UxgD,
>
> Thank you for being a reviewer for NeurIPS2023, your service is invaluable to the community!
>
> The authors have already submitted their feedback and I noticed that you don't appear to have submitted a new round of comments.
>
> Could you examine rebuttals and other reviewers' comments, and open up discussions with the authors and other reviewers?
>
> Regards, Your AC

---

### Official Review · Reviewer_d53F · 2023-07-05

**Soundness:** 3 good
**Presentation:** 3 good
**Contribution:** 2 fair
**Rating:** 7
**Confidence:** 4

**Summary:**

This paper explores the notion of using random weight networks as a constraint during the training process for image restoration. This approach aims to encourage the network to learn more robust features and produce better results, addressing the limitations of traditional optimisation methods and deep learning-based methods. By incorporating the random weight network as a constraint, the authors validate the approach towards improving image restoration performance.

**Strengths:**

1. The authors provide sufficient theoretical insights behind the formulation of using a randomly initialised network as an auxiliary loss function during the optimisation process.

2. The ablation studies are elaborate and cover a wide variety of initialisation configurations and examine its effect on final restoration performance.

**Weaknesses:**

1. Experimental Setting section is repeated

2. The proposed approach is similar to [1, 2, 3] and without discussion on differences, the contribution of the proposed work is weak. Specifically identification of different distributions and its impact on final restoration performance.

3. The authors should discuss the impact of utilising multiple network architectures on the overall training period as well as memory requirements.

4. The impact of initialisation distribution should be discussed, which is missing. Furthermore in the qualitative results the authors should also provide corresponding input and ground truth images for easier evaluation.

5. While the authors evaluated the impact of network structures by replacing the CNN with transformer architectures in ablation. Other configurations such as using transformer based restoration networks and implications of using a lightweight optimisation network aren't considered. These ablations are necessary to identify the overall implication of using different strategies during optimisation.


[1] Gallicchio, Claudio, and Simone Scardapane. "Deep randomized neural networks." Recent Trends in Learning From Data: Tutorials from the INNS Big Data and Deep Learning Conference (INNSBDDL2019). Springer International Publishing, 2020.

[2] Herrera, Calypso, et al. "Optimal stopping via randomized neural networks." arXiv preprint arXiv:2104.13669 (2021).

[3] Tarvainen, Antti, and Harri Valpola. "Mean teachers are better role models: Weight-averaged consistency targets improve semi-supervised deep learning results." Advances in neural information processing systems 30 (2017).

**Questions:**

Kindly address the comments raised in weakness section

**Limitations:**

The authors have addressed the limitations arising from space but not the limitation of their methodology, which was the original objective.

---

> ### Author Rebuttal · Authors · 2023-08-08
>
> **1, typos.**
>
> Thanks for highlighting the issue! We'll thoroughly review the entire paper and correct all typos and grammar errors.
>
> **2, differences to other works.**
>
> 1) Our method centers on designing the loss function, initially employing a random weights network with a strict mathematical manifold as a loss constraint, inspired by Functional theory. In contrast, the suggested works aim to explore a suitable subnet within the random network, known as ticket theory, to function as the working network. This leads to a distinction: our proposed random network serves the loss function, while theirs serves as a feed-forward network. Theoretical assurance for our approach is straightforward due to its strict mathematical foundation.
>
> 2) As per your suggestion, we will incorporate a detailed discussion of the differences to enhance the content.
>
> 3) Our work delves into identifying various distributions, as discussed in Tables 8 and 9.
>
> **3, memory requirements.**
>
> Initially, we explore the effects of employing diverse network architectures on training duration in our supplementary materials. Moreover, all random weights networks in our study are lightweight; specifics are outlined in the supplementary materials. Consequently, the utilization of multiple network architectures introduces minimal and inconsequential memory demands throughout the training process.
>
> **4, initialization distribution.**
>
> Initially, we address the initialization distribution impact, as demonstrated in Table 8 and Table 9 with analysis in the ablation studies section. Furthermore, we'll heed your advice to include input and ground truth images for qualitative results.
>
> **5, network structures.**
>
> All our random weights networks are lightweight; see supplementary materials for detailed configurations. For model depth, specific experiments are in Table 6 and Table 7. In the transformed-based approach, we validate our claim using the INNformer model and its results. Additionally, due to time constraints, we tested the effectiveness of SWINIR for image denoising, as shown in the subsequent table.
>
>
>
> |dataset|SWINIR  |    +Taylor  |   +INN    |  +Zerofilter |
> |----|----|----|----|----|
> |Set12      |     31.01   |    31.22  |   31.29    | 31.27 |
> |BSD     |       29.50   |    29.63   |   29.67  |   29.65 |

---

> > ### Comment · Reviewer_d53F · 2023-08-21
> > **Response to Authors**
> >
> > I thank the authors for their detailed reply which addressed my concerns. Thus I upgrade my rating for this work.

---

### Decision · Program_Chairs · 2023-09-21

**Decision:**

Accept (poster)

**Comment:**

The final ratings received for this paper were A, SA, SA, BR, indicating an overall positive trend in the evaluations. The majority of reviewers expressed positive opinions about the theoretical explanations in the work and commended the credibility of the experimental validation, and they noted that their concerns were appropriately addressed by the authors in their responses. As for the reviewer who tended to reject, there did not appear to be further comments after the author submitted the rebuttal. Nevertheless, the previous concerns raised by this reviewer regarding the technical feasibility and the breadth of experimental validation still need careful consideration. Above all, this work demonstrates an overall high quality, but there are some issues in the manuscript that require the author's careful consideration and improvement. Therefore, I believe that with some revisions, this work can meet the acceptance standards, and I have decided to accept it.